# Estrogen Deficiency Potentiates Thioacetamide-Induced Hepatic Fibrosis in Sprague-Dawley Rats

**DOI:** 10.3390/ijms20153709

**Published:** 2019-07-29

**Authors:** Yong Hee Lee, Ji Yeon Son, Kyeong Seok Kim, Yoo Jung Park, Hae Ri Kim, Jae Hyeon Park, Kyu-Bong Kim, Kwang Youl Lee, Keon Wook Kang, In Su Kim, Sam Kacew, Byung Mu Lee, Hyung Sik Kim

**Affiliations:** 1School of Pharmacy, Sungkyunkwan University, Suwon 16419, Korea; 2College of Pharmacy, Dankook University, Cheonan 31116, Korea; 3College of Pharmacy & Research Institute of Drug Development, Chonnam National University, Gwangju 61186, Korea; 4College of Pharmacy, Seoul National University, Seoul 08826, Korea; 5McLaughlin Centre for Population Health Risk Assessment, University of Ottawa, Ottawa, ON K1N 6N5, Canada

**Keywords:** liver fibrosis, thioacetamide, estrogen deficiency, α-SMA, collagen I

## Abstract

Hepatic fibrosis is characterized by persistent deposition of extracellular matrix proteins and occurs in chronic liver diseases. The aim of the present study is to investigate whether estrogen deficiency (ED) potentiates hepatic fibrosis in a thioacetamide (TAA)-treated rat model. Fibrosis was induced via intraperitoneal injection (i.p.) of TAA (150 mg/kg/day) for four weeks in ovariectomized (OVX) female, sham-operated female, or male rats. In TAA-treated OVX rats, the activities of serum alanine aminotransferase (ALT), aspartate aminotransferase (AST), alkaline phosphatase (ALP), and γ-glutamyl transferase (GGT) were significantly increased compared to those in TAA-treated sham-operated OVX rats or TAA-treated male rats. Furthermore, α-smooth muscle actin (α-SMA) expression was significantly increased compared to that in TAA-treated sham-operated rats. This was accompanied by the appearance of fibrosis biomarkers including vimentin, collagen-I, and hydroxyproline, in the liver of TAA-treated OVX rats. In addition, ED markedly reduced total glutathione (GSH) levels, as well as catalase (CAT) and superoxide dismutase (SOD) activity in TAA-treated OVX rats. In contrast, hepatic malondialdehyde (MDA) levels were elevated in TAA-treated OVX rats. Apoptosis significantly increased in TAA-treated OVX rats, as reflected by elevated p53, Bcl-2, and cleaved caspase 3 levels. Significant increases in interleukin-6 (IL-6) and tumor necrosis factor-α (TNF-α) concentrations were exhibited in TAA-treated OVX rats, and this further aggravated fibrosis through the transforming growth factor-β (TGF-β)/Smad pathway. Our data suggest that ED potentiates TAA-induced oxidative damage in the liver, suggesting that ED may enhance the severity of hepatic fibrosis in menopausal women.

## 1. Introduction

Liver fibrosis is characterized by persistent deposition of extracellular matrix (ECM) proteins and it occurs in most types of chronic liver disease, including fibrosis and cirrhosis [1]. Cirrhosis, the twelfth leading cause of death worldwide, was responsible for 1 million deaths in 2010 [2]. Although the main causes of hepatic cirrhosis may be associated with increased alcohol consumption and obesity, poor lifestyle is also considered a risk factor. The gender difference in prevalence of cirrhosis is increasing, with men showing a higher incidence than women [3]. Further, the prevalence of non-alcoholic fatty liver disease (NAFLD) is lower in pre-menopausal women during the reproductive period than in men, but comparable in post-menopausal women and men [4]. Therefore, post-menopausal women might exhibit more severe hepatic disease progression and an elevated incidence of hepatic fibrosis. These findings may be explained by the protective effects of estrogen against hepatic fibrogenesis. It has been clearly demonstrated that the protective effects of hormone replacement therapy against fibrogenesis in hepatitis virus related liver disease appears to be mediated by estrogen [5]. However, exactly how gender and reproductive state impact the severity of fibrosis in patients with NAFLD remains largely unknown [2,6,7].

In fibrosis conditions, hepatocyte necrosis or apoptosis occurs in hepatic tissue through Kupffer cell activation, eventually leading to cirrhosis [8]. Activated Kupffer cells secrete inflammatory cytokines, such as transforming growth factor-β (TGF-β), to activate hepatic stellate cells (HSCs). Activated HSCs produce fibrosis through excessive deposition of extracellular matrix (ECM) proteins, such as collagen and vimentin [9,10,11]. TGF-β, an important pro-fibrotic cytokine, plays a key role in ECM synthesis via the TGF-β/Smad signaling pathway [12,13]. TGF-β uses intracellular signaling pathways other than Smad to regulate various cellular functions [14]. The phosphoinositide 3-kinase (PI3K)/protein kinase B (Akt) pathway is another non-Smad pathway contributing to TGF-β-induced epithelial-mesenchymal transition (EMT) in liver fibrosis [13].

Thioacetamide (TAA) is a fungicide widely used to prevent decay in fruits, although it is classified as a group 2A human carcinogen [14]. TAA is widely employed to induce fibrosis in animal models because it results in morphological and biochemical characteristics similar to those observed in human liver cirrhosis [15,16]. TAA produces damage not only in the liver, but also other organs [17], and is metabolized by hepatic cytochrome (CYP) 450 to generate TAA S-dioxide [18]. This active product induces oxidative stress and production of TGF-β and reactive oxygen species (ROS) [19]. TAA-induced liver fibrosis is established when HSCs are activated through lipid peroxide formation, cytotoxicity, and mitochondrial injury. Taken together, these events may lead to apoptosis, necrosis, formation of cholangiocarcinoma, and excessive accumulation of ECM proteins [20].

Epidemiological studies have shown that female hormones play an important role in preventing chronic inflammatory diseases [21,22]. Estrogen is a steroid hormone that regulates differentiation and growth in various mammalian tissues, including the breast, vagina, endometrium, uterus, and liver [23]. Estrogen promotes DNA synthesis in hepatocytes, while inhibiting the release of ROS and inflammatory cytokines from Kupffer cells [24]. Kupffer cells secrete inflammatory substances such as TGF-β, tumor necrosis factor-α (TNF-α), and interleukin (IL)-6 to activate HSCs [25]. In general, estrogen interacts with the estrogen receptor (ER) directly or indirectly to affect cell growth and proliferation [26]. ERα and ERβ display high specificity for target tissues, and ERα is highly expressed in the liver, epididymis, prostate stroma, breast, and uterus [27,28]. ERα is reportedly expressed in up to 40% of hepatocytes in the human liver [29]. In contrast, ERβ is highly expressed in the lungs, testis, prostate epithelium, bone marrow, and brain [30]. Furthermore, ERα is highly expressed in mouse hepatic tissue [31,32]. Therefore, the liver is a target organ for estrogen [33,34]. A previous study demonstrated that hepatic expression of ER was negatively correlated with α-smooth muscle actin (SMA) and TGF-β expression, which were higher in the tissues of male patients suffering from liver fibrosis than in female patients [35].

Although estrogen protects against hepatic fibrosis by suppressing Kupffer cell activity, the molecular mechanisms underlying the effects of estrogen deficiency (ED) against liver diseases remain unclear. Therefore, the aim of our study was to investigate whether ED can increase the severity of hepatic fibrosis in a TAA-induced animal model. Furthermore, the precise mechanism for increased susceptibility of females to chemical-induced liver diseases is not completely understood. The present study also compared the potency of hepatic fibrosis according to gender differences.

## 2. Results

### 2.1. Effects of ED on TAA-Mediated Body and Liver Weight Changes

Liver fibrosis was induced by injection of TAA (150 mg/kg/day) for four weeks (Figure 1). The general status of the animals in the male control or sham-operated control groups was excellent, showing gradual increases in body weight gain. However, the OVX/TAA, sham-operated TAA, and male TAA groups exhibited significant reductions in body weight gain (Table 1). In the OVX group, uterine weight was significantly lower than in the sham-operated control group (Table 1). Liver weight was higher in the OVX/TAA group than in the sham/TAA group. Similarly, liver weight was increased in the male TAA group compared to the male control (Table 1).

### 2.2. Effect of ED on TAA-Induced Liver Damage

Serum alanine aminotransferase (ALT) and aspartate aminotransferase (AST) activities significantly increased in the TAA-treated OVX group compared to the sham/TAA group (Figure 2A). In addition, AST, ALT, alkaline phosphatase (ALP), and γ-glutamyl transferase (γ-GGT) activities were significantly elevated in the male TAA group compared to the male control. These results indicate that the severity of TAA-induced hepatic damage was greater in ED or male rats than in normal female rats. Circulating TNF-α and IL-6 levels were higher in the OVX/TAA group than in the sham/TAA group (Figure 2B). Therefore, serum TNF-α and IL-6 play an important role in the development of TAA-induced hepatic fibrosis.

Histological examination of the liver was performed using a light microscope. Hematoxylin and eosin (H&E)-stained liver sections from the control groups showed clear portal areas containing elements of the hepatic triad, a small bile duct along with lymphatic vessels, and a small amount of connective tissue. Liver cells were arranged in plates or cords and radiated from the central venue regions. However, significant increases in infiltration of inflammatory or necrotic cells were clearly exhibited in the OVX/TAA and male TAA groups. However, these damaged cell populations were reduced in the sham/TAA group, indicating that ED increased the severity of TAA-induced hepatic damage (Figure 2C).

### 2.3. Effect of ED on ER Expression in the Livers of TAA-Treated Rats

Hepatic ERα expression was significantly reduced in the OVX/TAA or male TAA groups (Figure 3A). In addition, ERα expression was also lower in the sham/TAA group than the sham control group. These results were confirmed again using IHC staining. As shown in Figure 3B, ERα is mainly expressed in normal hepatocyte nuclei. ERα expression was high in the control groups, but significantly lower in the OVX/TAA and male TAA groups. Therefore, we suggest that ERα expression plays an important role in protecting against hepatic damage.

### 2.4. Influence of ED on Hepatic Antioxidant Enzyme Activities

Levels of the antioxidant enzymes SOD and CAT, as well as levels of GSH and MDA, were determined in rat hepatic tissue. Total SOD and CAT enzyme activities were significantly reduced in the OVX/TAA or male TAA groups, compared to the control groups. Similarly, GSH concentration was also lower in OVX/TAA and male TAA groups compared to the sham/TAA group. In particular, MDA levels were significantly elevated in the OVX/TAA or male TAA groups compared to the sham/TAA group (Figure 4A).

### 2.5. Effect of ED on Hepatocyte Apoptosis

Apoptosis-related protein expression was determined in the liver. The protein expression of p53 and BAX levels were higher in OVX/TAA and male TAA groups, compared to the control groups. However, the expression of Bcl-2 protein was absolutely inhibited in all the TAA-treated groups compared to controls. Cleaved caspase-3, one of the last active proteins required for apoptosis, was markedly increased in the OVX/TAA and male TAA groups compared to the control groups (Figure 4B). In particular, the expression of apoptosis-related proteins was significantly increased in the OVX/TAA and male TAA groups compared to the sham/TAA group. Few TUNEL-positive cells (hepatocytes and non-parenchymal cells) were detected in the livers of the control groups; however, the number of TUNEL-positive cells was significantly elevated in the OVA/TAA and male TAA groups compared to the sham/TAA group (Figure 4C). TUNEL-positive apoptotic cells were scattered in the hepatic lobules of TAA-treated rats.

### 2.6. Influence of ED on Liver Fibrosis

The protein expression of α-SMA, collagen I, and vimentin was measured in the rat livers by western blotting analysis. The hepatic expression of α-SMA, collagen I, and vimentin proteins was increased in the OVX/TAA and male TAA groups compared to the sham/TAA and the control groups (Figure 5A). Hydroxyproline content in the serum and tissue was elevated in the OVX/TAA group compared to the sham/TAA group, indicating that hepatic fibrosis was successfully induced in these rats (Figure 5B). Hydroxyproline levels rose according to the severity of liver histological damage. We confirmed collagen deposition using immunohistochemical (IHC) findings. As shown in Figure 5C, we compared the expression of α-SMA, a marker of HSC activation, in liver tissues from both the TAA-treated and control groups. The OVX/TAA and male TAA groups showed enhanced expression of α-SMA, with α-SMA-positive HSCs observed in the damaged areas. Liver sections were then stained with MT to detect collagen deposition, revealing increased collagen accumulation (blue) in the OVX/TAA and male TAA groups compared to the sham/TAA and control groups (Figure 5D).

### 2.7. TGF-β1/Smad Signaling-Mediated Effects of ED on Liver Fibrosis

Smad4 protein expression was upregulated in the OVX/TAA and male TAA groups compared to the sham/TAA and control groups (Figure 6A). However, expression of the Smad2/3 protein did not differ between the TAA and control groups; thus, the Smad2/3 pathways were not involved in TGF-β1-induced hepatic fibrosis. As shown in Figure 6B, TGF-β expression markedly increased in the OVX/TAA and male TAA groups compared to the sham/TAA group. In particular, TGF-β1 was strongly localized in the damaged hepatic tissues. Thus, TGF-β signaling is involved in the development of hepatic fibrosis. We further investigated whether the TGF-β1/non-Smad, PI3K, and AKT pathways were involved in the progression of TAA-induced hepatic fibrosis. p-PI3K and p-Akt protein expression was highly elevated in the liver of OVX/TAA and male TAA groups compared with sham/TAA or control groups (Figure 6C). This finding was inversely correlated with PTEN protein expression. p-PTEN expression was dramatically reduced in the OVX/TAA and male TAA groups (Figure 6C).

## 3. Discussion

Postmenopausal women exhibit a higher incidence of hepatic fibrosis than premenopausal women and men. Thus, the sex hormone estrogen is thought to play a critical role in preventing hepatic fibrosis [6,7,36]. Although numerous investigators have suggested the protective effects of estrogen against hepatic fibrosis [23,36], how ED status may increase the severity of chemical-induced hepatic fibrosis remains unclear. Our findings clearly demonstrated that OVX female rats potentiated the severity of TAA-induced hepatic fibrosis in sham control rats. Further, the potency of hepatic damage exhibited in OVX rats was very similar to that shown in male rats. In the hepatic tissues, ERα protein expression was dramatically reduced, promoting secretion of IL-6 and TNF-α in the OVX/TAA groups. Therefore, our results suggest that hepatic fibrosis progresses in postmenopausal women suffering from ED due to a reduction in ERα protein expression in the liver.

TAA-induced hepatic fibrosis is accompanied by increasing inflammation and oxidative stress in rats [37]. Thus, this model is appropriate for investigating the effects of ED in OVX female rats. OVX-induced ED results in decreased uterus weight [38], and, in this study, uterine weight was significantly reduced in OVX rats. In addition, body weight loss and liver weight gain occurred in TAA-treated animals [19]. In this study, liver weight increased in the OVX/TAA group compared to the sham/TAA group, suggesting that the accumulation of ECM proteins increased the liver weight in the OVX/TAA group, and ED further increased the severity of ECM accumulation, leading to the progression of hepatic fibrosis. To assess hepatic damage, serum ALT, AST, ALP, and γ-GTT activity was measured. These enzyme activities were significantly elevated in the TAA-treated rats, and the OVX/TAA group showed further exacerbated hepatic damage compared with sham/TAA. Further, cell necrosis and infiltration were also enhanced in the OVX/TAA group, according to histopathological findings. Therefore, ED accelerated hepatic dysfunction via hepatocytes or NPC damage in TAA-treated rats. A previous study indicated that estrogen blocks HSC activity by inhibiting the secretion of cytokines that protect against liver fibrosis [24] and interacts with the ER to regulate the transcriptional regulation of ER-mediated target genes [26]. Decreased ERα protein expression in the liver is an independent predictor of fibrosis [35,36]. Immunological staining and western blotting analysis showed that expression of ERα protein levels was dramatically reduced in all TAA-treated rats, and ED further diminished ERα protein expression. Therefore, ERα may play an important role in inhibiting hepatocyte injury and preventing liver fibrosis.

ROS affect various biological processes, including cell differentiation, gene expression, and cytokine responses. Parola and Robino [37] noted that oxidative stress caused by TAA plays an important role in inducing liver fibrosis. ROS produce liver injury by damaging lipids, proteins, and DNA in hepatocytes [39]. Antioxidant protection systems counteract the effects of ROS and inhibit severe liver injury [40]. SOD and CAT activities, as well as GSH levels, are indicators of oxidative stress. MDA is an indicator of lipid peroxidation. Lipid peroxidation changes the properties of biological membranes and results in severe cell damage, and this phenomenon is postulated to play a significant role in the pathogenesis of hepatic injury. Lipid peroxidation, free-radical-mediated processes, and certain lipid peroxidation products induce genetic overexpression of fibrogenic cytokines and increase collagen synthesis. ROS and MDA may stimulate collagen synthesis and initiate HSC activation [37,41]. In the present study, our data showed that OVX/TAA groups significantly reduced antioxidant enzyme activities and enhanced lipid peroxidation. Thus, ED may accelerate hepatic fibrosis by activating TAA-mediated oxidative stress.

Chronic liver injury is characterized by apoptosis, regeneration, and progressive fibrosis. Hepatocyte apoptosis is prominent in hepatic pathology and is a key step in most forms of liver injury [42,43]. In addition, several proinflammatory cytokines, including IL-1β, IL-6, and TNF-α, induce unresolved hepatocytic inflammation. Our data indicated the proinflammatory cytokine levels significantly increased in OVX/TAA rats compared to the sham control. Progressive hepatic fibrosis was, therefore, accompanied by an increase of proinflammatory cytokines. Further, TGF-β is secreted by immune cells at the site of injury for phagocytosis of apoptotic bodies, thus fueling inflammatory and fibrogenic reactions [44]. In hepatic fibrosis animal models, estradiol treatment markedly suppressed early apoptosis and hepatic fibrosis, reduced collagen content and α-SMA expression, and induced Bcl-2 expression [45,46,47]. In our ED OVX rats, p53, BAX, and cleaved caspase-3 expression were significantly increased compared to the sham/TAA rats. Therefore, ED accelerated apoptosis in the progression of hepatic fibrosis, whereas high levels of estrogen might diminish cell differentiation and enhance cell proliferation. Hepatocyte-specific inhibition of antiapoptotic proteins results in increased hepatocyte apoptosis, elevated serum ALT levels, and hepatic fibrosis development [48,49]. These findings suggest that hepatocytes are constitutively subjected to proapoptotic stress, even in the absence of disease, which is consistent with the presence of ALT in the serum of healthy subjects [49]. Apoptosis is, therefore, recognized as a prominent feature of liver injury.

Liver fibrosis is a wound-healing response following liver injury. During hepatic fibrosis, activated HSCs undergo continuous proliferation, as reflected by α-SMA activation [50]. Excessive matrix proteins are secreted by HSCs, and ECM protein accumulation is a key factor in liver fibrosis [1,51,52]. In this study, fibrosis indicators such as α-SMA, vimentin, and collagen I were highly expressed in the OVX/TAA and male TAA groups compared to the sham/TAA group. Fibrosis severity was also assessed via measurement of 4-hydroxyproline in the liver or serum, as previously reported [53]. Hydroxyproline content was significantly elevated in the sham/TAA and male TAA rats, and further increased in the OVX/TAA rats with ED. These results were clearly exhibited by MT and IHC staining to show that OVX accelerated ECM accumulation and myoblast differentiation in the liver of TAA-treated rats. The TGF-β1 signaling pathway has been shown to be an important pro-inflammatory cytokine pathway because of its influence on HSC activity [12]. TGF-β1 activates HSCs by the Smad pathway and promotes ECM protein production [9,11]. Our data showed that ED potentiated liver fibrosis through the TGF-β1/Smad pathway, and that TGF-β1 regulated liver fibrosis through the PI3K/Akt pathways.

In summary, ED activated hepatocyte apoptosis by increasing ROS generation and pro-inflammatory cytokine secretion, ultimately promoting hepatic fibrosis. Our data suggest that ED potentiates the severity of TAA-induced hepatic fibrosis, indicating that postmenopausal status may enhance the severity of hepatic fibrosis in menopausal women. However, further studies are needed to elucidate the basic mechanisms underlying the role of ED or ER in the acceleration of hepatic fibrosis.

## 4. Materials and Methods

### 4.1. Chemicals and Reagents

Thioacetamide (TAA) was purchased from Sigma-Aldrich (St. Louis, MO, USA) and dissolved in normal saline (0.9% sodium chloride). Primary antibodies against α-SMA, collagen-I, Smad4, and B-cell lymphoma 2 (Bcl-2)-associated X protein (BAX) were purchased from Abcam (Cambridge, UK). ER-α, Smad7, Bcl-2, p53, and β-actin were obtained from Santa Cruz Biotechnology (Dallas, TX, USA). Vimentin, p-Smad2/3, Smad2/3, cleaved caspase 3, p-PI3K, p-Akt, phosphatase and tensin homolog (PTEN), and p-PTEN were purchased from Cell Signaling Technology (Danvers, MA, USA). Assay kits used to measure aspartate aminotransferase (AST), alanine aminotransferase (ALT), alkaline phosphatase (ALP), gamma-glutamyl transferase (γ-GGT), malondialdehyde (MDA), and catalase (CAT) were purchased from Abcam (Cambridge, UK). Total glutathione (GSH) assay kits were purchased from Enzo Life Sciences (New York, NY, USA). SOD assay kits were purchased from Cayman Chemical (Ann Arbor, MI, USA). Hydroxyproline assay kits were purchased from Cell Biolabs, Inc. (San Diego, CA, USA). All other chemicals used in this study were purchased from Sigma-Aldrich (St. Louis, MO, USA).

### 4.2. Experimental Design

Six-week-old Sprague-Dawley male (210 ± 5 g) and female rats (165 ± 5 g) were purchased from Charles River Laboratory Animal Resources (OrientBio, Sungnam, Korea) and maintained in a specific pathogen-free (SPF) room with 12 h light/dark cycles (ambient air temperature, 23 ± 0.5 °C; relative humidity, 55% ± 1%). Prior to the experiments, the animals were examined for any overt signs of illness, and only healthy animals were selected for further study. Tap water and rodent chow (OrientBio, Sungnam, Korea) were provided ad libitum. All animals were acclimated for 1 week, and then we randomly assigned female rats to the ovariectomy (OVX) group. OVX was performed under anesthesia with isoflurane, and the ovaries were dissected from the ovarian capsules. The experimental protocol used in this research was approved by the Sungkyunkwan University Institutional Animal Care and Use Committee (A20180420-2862, 20 April 2018).

TAA was dissolved in normal saline (0.9 % sodium chloride), and intraperitoneally (i.p.) injected daily for 4 weeks to induce liver fibrosis according to a previous study [14]. Rats were divided into five groups (*n* = 6 per group); sham control group (normal saline); TAA (150 mg/kg/day)-treated OVX group, TAA (150 mg/kg/day)-treated sham group, male control group; and male TAA (150 mg/kg/day)-treated group. At the end of the experiment, all animals were sacrificed after 24 h of fasting. Livers were collected for histological and other analyses. Sera were obtained via centrifugation at 4 °C, and frozen until further use.

### 4.3. Serum Biochemical Analysis

Activities of serum ALT, AST, ALP, and γ-GGT were analyzed via VetScan analyzer (Abaxis, Union City, CA, USA). TNF-α and IL-6 levels were measured via enzyme-linked immunosorbent assay (ELISA) kits, following the manufacturer’s instructions (Thermo Fisher Scientific, Waltham, MA, USA). Briefly, diluted serum samples were added in duplicate to 96-well plates coated with assay buffer and biotin conjugate and incubated at 25 °C for 2 h. After washing thrice with buffer, streptavidin-horseradish peroxidase (HRP) was added to each well, and the plate was incubated at 25 °C for 1 h. After further washes, the plate was incubated with 3,3′,5,5′-tetramethylbenzidine (TMB) substrate at 25 °C for 10 min. The reaction was stopped by adding stop solution. Optical density was determined at 450 nm using a Microplate spectrophotometer (Molecular Devices). Sample concentration was determined using a standard curve.

### 4.4. Histopathological Examination and Masson’s Trichrome (MT) Staining

Liver tissues were fixed in 10% neutral buffered formalin for 12 h and dehydrated with 70% ethanol. To detect collagen fibers, paraffin-embedded liver sections were cut to 5 μm thickness. Liver slide sections were stained with hematoxylin and eosin (H&E) to evaluate histopathologic changes, and with Masson’s trichrome (MT) stain to evaluate fibrosis. Tissue was visualized using a confocal laser-scanning microscope (Nanoscope Systems, Daejeon, Korea).

### 4.5. Glutathione (GSH) Content Determination

GSH content was measured using a kit (Enzo Life Sciences), according to the manufacturer’s instructions. Liver tissues were washed with phosphate-buffered saline (PBS), homogenized with 5% metaphosphoric acid, and centrifuged (12,000 × *g*) for 15 min at 4 °C. The supernatant was transferred to a sterile tube and the remaining sample was stored at 4 °C. Each specimen was added to the hepatic tissue sample and absorbance was measured at 405 nm. GSH content was expressed as nmol/mg of protein and quantified using a standard curve.

### 4.6. Assay for Antioxidant Enzyme Activities

Catalase (CAT) activity was measured using a kit (Abcam), according to the manufacturer’s instructions. Liver tissues were washed with PBS, homogenized with ice-cold assay buffer, and centrifuged (10,000 × *g*) for 15 min at 4 °C. The supernatant was transferred to a sterile tube and stored at 4 °C. Each specimen was added to hepatic tissue samples, and absorbance was measured at 570 nm. Superoxide dismutase (SOD) activity was determined using a kit (Cayman Chemical), according to the manufacturer’s instructions. Liver tissues were washed with PBS, and tissues were homogenized in HEPES buffer containing ethylene glycol tetraacetic acid (EGTA), mannitol, and sucrose at pH 7.2, and centrifuged at 1500 × g for 5 min at 4 °C. Each sample was added to hepatic tissue samples, and absorbance was measured at 440 nm. SOD activity was expressed as U/mL and quantified using a standard curve.

### 4.7. Malondialdehyde Assay

Malondehydehyde (MDA) concentrations were calculated using a TBARS assay kit (cat. no. KGE013; R&D Systems, Inc.), in accordance with the manufacturer’s instructions, and normalized to protein concentration. In brief, equal volumes (100 μL) of sample and sodium dodecyl sulfate were added to a 5 mL conical vial. After vortex mixing, the samples were mixed with 0.4 mL of 1% thiobarbituric acid (TBA) in 50 mm NaOH and 0.2 mL (20%) H_3_PO_4_. The mixture was heated to 100 °C for 15 min. After 10 min incubation on ice, the vials were centrifuged at 1600 × *g* for 10 min at 4 °C. The samples (100 µL) were loaded onto 96-well assay plates, and the absorbance of each well was measured at 540 nm using a microplate reader.

### 4.8. Western Blotting Analysis

Frozen liver samples were homogenized in PRO-PREPTM^TM^ protein extract solution (iNtRON, Seongnam, Korea) and centrifuged at 12,000 × g for 10 min. Protein concentrations were determined using a kit (Bio-Rad, Hercules, CA, USA), according to the manufacturer’s instructions. Equal amounts of protein were loaded on 6%–15% sodium dodecyl sulfate-polyacrylamide gels. Following electrophoresis, gels were transferred to a polyvinylidene difluoride (PVDF) membrane (Millipore, Burlington, MA, USA). The membrane was incubated for 60 min in TRIS NaCl Tween^®^ 20 (TNT) buffer (10 mM Tris-Cl, pH 7.6, 100 mM NaCl, and 0.5% Tween^®^ 20) containing 5% skimmed milk. Membranes were incubated with various primary antibodies (1:1000) against α-SMA, vimentin, collagen-I, ER-α, p-Smad2/3, Smad2/3, Smad4, Smad7, Bcl-2, BAX, caspase 3, cleaved caspase 3, p53, p-PI3K, PI3K, p-Akt, Akt, p-PTEN, PTEN, and β-actin at 4 °C for 12 h. The membranes were washed for 60 min with TNT buffer, incubated with HRP-conjugated anti-mouse immunoglobulin G (IgG, 1:20,000) or anti-rabbit IgG (1:20,000) antibodies for 60 min at 25 °C, and washed for 60 min with TNT buffer. The blots were developed using an enhanced chemiluminescence (ECL)-plus kit (Amersham Biosciences, Little Chalfont, UK).

### 4.9. Hydroxyproline Assay

The collagen level was determined by estimating the hydroxyproline level, an amino acid characteristic of collagen. Hydroxyproline content in serum samples was determined using an ELISA kit (Cell Biolabs), following the manufacturer’s instructions. In a 2 mL sterilized tube, 100 μL serum and 500 μL HCl (2N) were mixed to allow hydrolysis. Samples were boiled for 3 h at 120 °C, cooled on ice for 5 min, and filtered using a 0.45 μm PVDF syringe filter. After centrifugation at 10,000 × g, only the supernatant was transferred to a new sterile tube. Absorbance was detected using a Multiplate Spectrometer (Ultramark, Bio-Rad) at 560 nm. A portion of the liver tissue (200 mg) was homogenized in 10 volumes of 0.5 mol/L potassium phosphate buffer, and hydroxyproline content was measured as described above.

### 4.10. Immunohistochemical Examination

Liver sections were treated with xylene and ethanol, and boiled in sodium citrate buffer for 20 min. Tissue sections were treated with 5% hydrogen peroxide (H_2_O_2_) for 10 min to deactivate endogenous peroxidase. Each slide was incubated with α-SMA (1:200), TGF-β (1:300), and ER-α (1:500) antibodies at 4 °C for 24 h. Next, the slides were treated with secondary anti-rabbit antibodies (Vector Labs, Burlingame, CA, USA) at room temperature for 30 min and then with Vectastan ABC reagent (Vector Labs) at room temperature for a further 30 min. After completion of the DAB+ substrate system (Agilent, Santa Clara, CA, USA) and hematoxylin reactions (Agilent), the slides were treated with ethanol and xylene, and then fixed in a cover glass using a mounting solution. The slides were visualized at 200× magnification using a confocal K1-Fluo microscope (Nanoscope Systems).

### 4.11. TUNEL Assay

A DeadEnd™ Colorimetric TUNEL System (Promega, Madison, WI, USA) was used for the assay. Liver slides were treated with xylene and ethanol, NaCl, 4% paraformaldehyde, and proteinase K, followed by treatment with rTdT at 37 °C for 2 h. Hydrogen peroxide, HRP, DAB, and hematoxylin (Agilent) were added to the slides, which were incubated for 30 min and then treated with ethanol and xylene. The slides were visualized at 200× magnification using a K1-Fluo microscope (Nanoscope Systems).

### 4.12. Statistical Analysis

Data are expressed as the mean ± S.D., and were analyzed by one-way analysis of variance (ANOVA) followed by the Tukey’s HSD post hoc test for multiple comparisons. Statistical analyses were performed using GraphPad Prism v5.0 (GraphPad Software, San Diego, CA, USA). Results were considered to be statistically significant when *p*-values were less than 0.05.

## Figures and Tables

**Figure 1 ijms-20-03709-f001:**
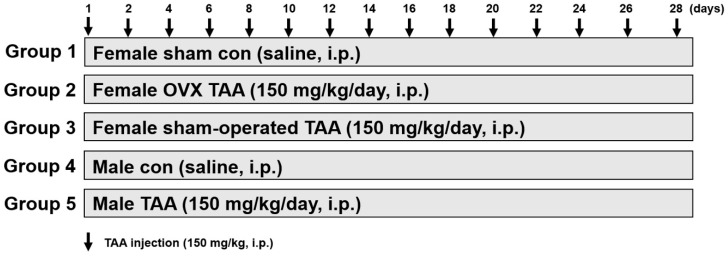
Experimental design for the liver fibrosis model. Liver fibrosis was induced by repeated injections of TAA (150 mg/kg/day) for 4 weeks in ovariectomized female or male rats. TAA = thioacetamide; i.p. = intraperitoneal injection; OVX = ovariectomized.

**Figure 2 ijms-20-03709-f002:**
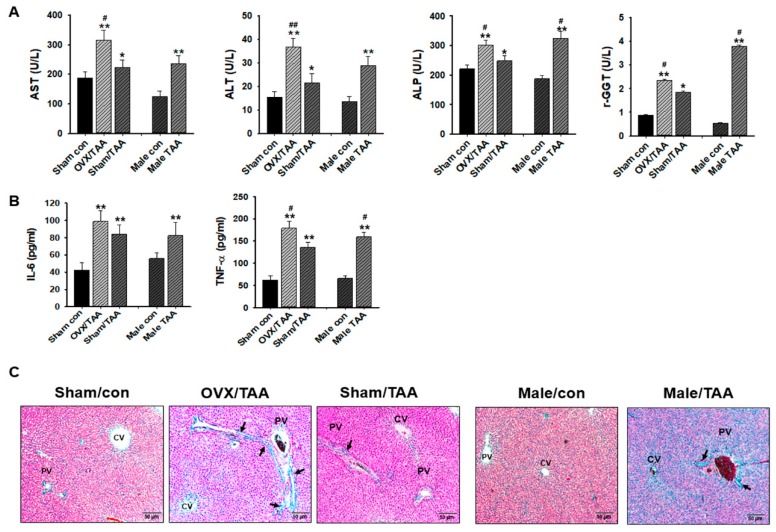
Effects of estrogen deficiency on hepatic damage in TAA-treated rats for 2 weeks. (**a**) Changes in serum aspartate aminotransferase (AST), alanine aminotransferase (ALT), alkaline phosphatase (ALP), and r-glutamyl transferase (r-GTT) activities in the serum of TAA-treated rats. Data are expressed as the mean ± S.D (6 animals/group). Statistical analysis was performed by one-way ANOVA followed by a Tukey’s HSD post hoc test for multiple comparisons. * *p* < 0.05 and ** *p* < 0.01 compared to control group; ^#^
*p* < 0.05 and ^##^
*p* < 0.01 compared to sham/TAA group. (**b**) Alterations in the pro-inflammatory cytokines interleukin 6 (IL-6) and tumor necrosis factor-alpha (TNF-α) levels in the serum of TAA-treated rats. Data are expressed as the mean ± S.D (6 animals/group). Statistical analysis was performed by one-way ANOVA followed by a Tukey’s HSD post hoc test for multiple comparisons. ** *p* < 0.01 compared to control group; ^#^
*p* < 0.05 compared to sham/TAA group. (**c**) Representative histology of hematoxylin and eosin (H&E) stained liver sections from experimental groups. TAA-induction for 10 days showed small nodules with degenerative hepatocytes, absence of sinusoid, an increase of fibrous tissue thickness (black arrows), and expansion of the portal tract with hepatic central vein and hepatic nodule increases (cirrhosis). CV, the central vein; PV, the portal vein. Original magnification: ×100.

**Figure 3 ijms-20-03709-f003:**
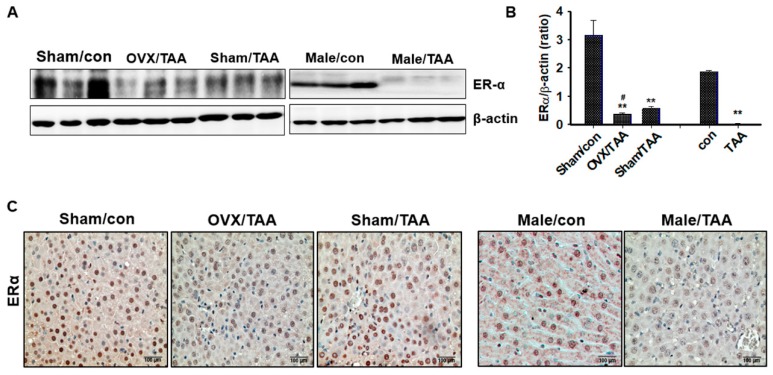
Effects of estrogen deficiency on estrogen receptor α (ERα) expression in livers of TAA-treated rats for 4 weeks. (**a**) Representative western blots of ERα expression in liver, using an experimental model of TAA-induced hepatic fibrosis. β-Actin expression was used as the loading control. (**b**) Protein bands were quantified by densitometric analysis and normalized to β-actin. Values are expressed as the mean ± S.D. ** *p* < 0.01 compared to control group; ^#^
*p* < 0.05 compared to sham/TAA group. (**c**) Representative immunohistochemical staining of ERα expression from experimental groups. Original magnification: ×100, scale bar: 100 μm. Arrow indicated the ERα positive cells.

**Figure 4 ijms-20-03709-f004:**
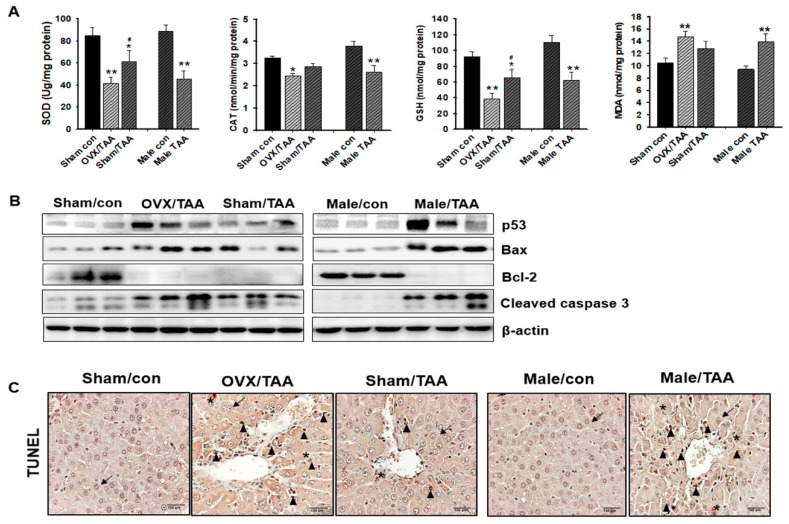
Effect of estrogen deficiency on antioxidant enzyme activity and apoptosis in the liver of TAA-treated rats. (**a**) Malondialdehyde (MDA), catalase (CAT), glutathione (GSH), and total superoxide dismutase (SOD) activities were measured in the liver of TAA-treated rats. Data are expressed as the mean ± S.D of duplicate experiments (6 animals/group). Statistical analysis was performed by one-way ANOVA followed by a Tukey’s HSD post hoc test for multiple comparisons. * *p* < 0.05 and ** *p* < 0.01 compared to control group; ^#^
*p* < 0.05 compared to OVX/TAA group. (**b**) Expression of p53, Bax, Bcl-2, and cleaved caspase-3 were measured by western blot analysis, using an experimental model of TAA-induced hepatic fibrosis. β-Actin expression was used as the loading control. The western blot results are representative of three separate experiments. (**c**) Apoptosis was determined by terminal deoxynucleotidyl transferase deoxyuridine triphosphate nick end labeling (TUNEL) staining of liver sections from TAA-treated rats. The arrowheads indicate TUNEL-positive hepatocytes and Stars indicate TUNEL-positive NPC. Arrows indicate hepatocytes. NPC, non-parenchymal cells. Magnification ×200.

**Figure 5 ijms-20-03709-f005:**
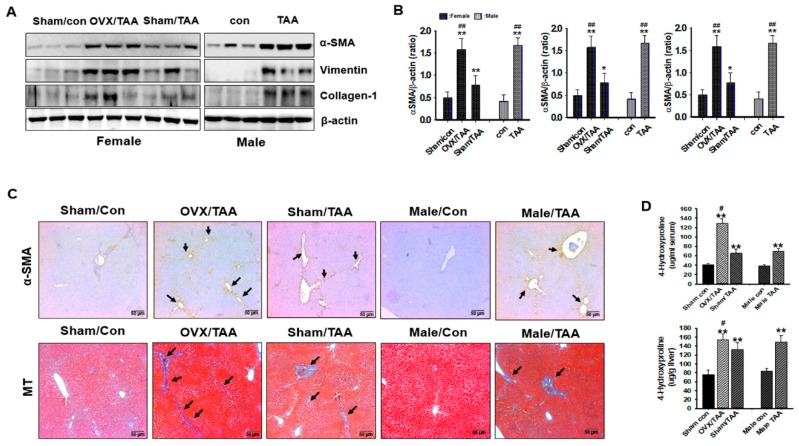
Effect of estrogen deficiency on extracellular matrix protein formation in the liver of TAA-treated rats. (**A**) Expression of α-SMA, collagen I, and vimentin were measured by western blot analysis using an experimental model of TAA-induced hepatic fibrosis. β-Actin expression was used as the loading control. The western blot results representative three separate experiments. (**B**) Protein bands were quantified by densitometric analysis and normalized to β-actin. Values are expressed as the mean ± S.D. * *p* < 0.05 and ** *p* < 0.01 compared to control group; ^##^
*p* < 0.01 compared to sham/TAA group. (**C**) Representative immunohistochemical staining of α-SMA and Masson’s trichrome-stained liver sections. Black arrows represent α-SMA and collagen accumulation. Original magnification: 40×, scale bar: 50 μm. (**D**) 4-Hydroxyproline contents in the serum and liver of TAA-treated rats. Data are expressed as the means ± S.D. of duplicate experiments (6 animals/group). Statistical analysis was performed by one-way ANOVA followed by a Tukey’s HSD post hoc test for multiple comparisons. ** *p* < 0.01 compared to control group; ^#^
*p* < 0.05 compared to sham/TAA group.

**Figure 6 ijms-20-03709-f006:**
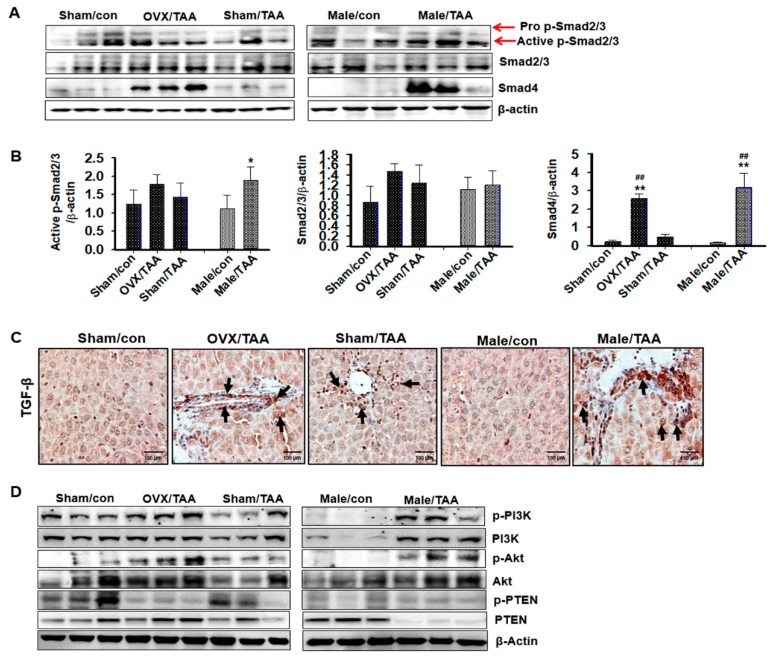
Effect of estrogen deficiency on hepatic fibrosis pathways in TAA-treated rats. (**a**) Expression of p-Smad2/3, Smad2/3, and Smad4 was measured by western blot analysis using an experimental model of TAA-induced hepatic fibrosis. β-Actin expression was used as the loading control. The western blot results representative three separate experiments. (**b**) Protein bands were quantified by densitometric analysis and normalized to β-actin. Values are expressed as the mean ± S.D. * *p* < 0.05 and ** *p* < 0.01 compared to control group; ^##^
*p* < 0.01 compared to sham/TAA group. (**c**) Representative immunohistochemical staining of TGF-β1 (Black arrows) in the liver. Original magnification: ×100 (**d**) Expression of phosphoinositol-3′-kinase (PI3K), p-PI3K, protein kinase B (Akt), p-Akt, phosphatase, and tensin homolog (PTEN) were measured by western blot analysis using an experimental model of TAA-induced hepatic fibrosis. β-Actin expression was used as the loading control. The western blot results representative three separate experiments.

**Table 1 ijms-20-03709-t001:** Effect of estrogen deficiency on body or organ weight changes in TAA-treated rats.

Groups	Body Weight (g)		Liver Weight (g)		Uterus Weight (g)	
	Initial (g)	Final (g)	Absolute (g)	Relative (%) ^a^	Absolute (g)	Relative (%) ^b^
Sham control	182.5 ± 2.5	245.6 ± 6.3	7.91 ± 1.05	3.21 ± 0.15	1.17 ± 0.05	0.48 ± 0.01
OVX TAA	181.9 ± 3.1	194.6 ± 5.7	8.69 ± 1.12 **	4.58 ± 0.21 **	0.11 ± 0.02 **	0.07 ± 0.02 **
Sham-operated TAA	184.1 ± 2.8	199.2 ± 4.7	7.92 ± 1.37	3.98 ± 0.27 ^##^	0.73 ± 0.04 ^##^	0.39 ± 0.05 ^##^
Male control	234.2 ± 3.8	373.8 ± 9.2	9.78 ± 1.52	2.74 ± 0.31	-	-
Male TAA	235.6 ± 4.1	245.4 ± 8.6	11.63 ± 1.48 **	4.73 ± 0.38 **	-	-

Data are expressed as the mean ± S.D (6 animals/group). Statistical analysis was performed by one-way ANOVA followed by a Tukey’s HSD post hoc test for multiple comparisons. ** *p* < 0.01 compared to control group; ^##^
*p* < 0.01 compared to OVX/TAA group. ^a^ % = [liver weight/body weight] × 100; ^b^ % = [Uterus weight/body weight] × 100.

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
