# Peer review of "Estrogen Deficiency Potentiates Thioacetamide-Induced Hepatic Fibrosis in Sprague-Dawley Rats"

_ijms, 2019, doi:10.3390/ijms20153709_

Round 1
Reviewer 1 Report
In the manuscript entitled “Estrogen deficiency potentiates thioacetamide-2 induced hepatic fibrosis in Sprague-Dawley rats”, Lee Y et al have described changes in specific pathways that may contribute to liver fibrosis during estrogen deficiency. The findings are interesting and potentially relevant to human studies. Other groups have performed studies in OVX rats with CCL4 injury and administration of estradiol to curb the deficiency. There are some concerns with data and overall discussion that need to be addressed:
Major concerns:
1. The control TAA male and the female with OVX+TAA are not very different.
The purpose of the study is to examine the effect of estrogen deficiency on TAA-induced liver fibrosis in females. However, the control male and TAA-male groups appear to show similar effect as seen in OVX females compared to their controls. So the question is how do the authors explain that OVX deficiency is a major player contributing to fibrosis in females? The authors need to justify better the use of the male control in all the experiments described.
2. Figure 5: alpha SMA should be high in the TAA female group and higher in OVX.
In figure 5, the authors show that OVX treated females exhibit higher SMA compared to controls. But the TAA alone female group should have higher SMA levels than control and the OVX+TAA should add on to the increase. In the figure shown, the TAA alone female group does not appear to exhibit any induction of SMA compared to control. In fact the Sham/Con appears to be higher than Sham/TAA in females.
Minor concerns:
Densitometries are required for all western blots because in some cases such as figure 5 (alpha-SMA) and figure 6 (SMAD3) , the changes show lot of variability.
Author Response
Reviewer #1
Comment 1. The control TAA male and the female with OVX+TAA are not very different. The purpose of the study is to examine the effect of estrogen deficiency on TAA-induced liver fibrosis in females. However, the control male and TAA-male groups appear to show similar effect as seen in OVX females compared to their controls. So the question is how do the authors explain that OVX deficiency is a major player contributing to fibrosis in females? The authors need to justify better the use of the male control in all the experiments described.
Response: Thank for your valuable comments. In generally, men are at a higher risk of having more severe fibrosis compared to women before menopause, while post-menopausal women have a similar severity of liver fibrosis compared to men. These findings may be explained by the severity of hepatic fibrosis in the state of estrogen deficiency. In particular, the pathophysiology of gender differences in the incidence, natural history, and outcomes of common liver diseases is incompletely understood. However, the precise mechanism for increased susceptibility or severity of postmenopausal women to chemical-induced fibrosis is not completely understood. The present study is based on the hypothesis that estrogen deficiency is the likely contributing factor for severity of hepatic fibrosis in females which is very similar to that of male. As your comment, we revised this part in text that estrogen deficiency (OVX/TAA) is also a major player contributing to hepatic fibrosis in females compared with sham/TAA. We added this revised part in Abstract and Introduction parts.
Comment 2. Figure 5: alpha SMA should be high in the TAA female group and higher in OVX. In figure 5, the authors show that OVX treated females exhibit higher aSMA compared to controls. But the sham/TAA alone female group should have higher SMA levels than control and the OVX+TAA should add on to the increase. In the figure shown, the sham/TAA alone female group does not appear to exhibit any induction of SMA compared to control. In fact, the Sham/Con appears to be higher than Sham/TAA in females.
Response: Thank you for your comments. We revised Fig. 5a in immunobloting analysis. These results are showed very similar trends in IHC. In Fig. 5c, liver fibrosis marker, alpha-SAM expression was much stronger in OVX+TAA group than sham/TAA group. In addition, sham/TAA group showed much higher expression of αSMA than that of shame/con group. These results are very similar to shown in the Masson’s trichrome-stained liver sections.
Comment 3. Densitometries are required for all western blots because in some cases such as figure 5 (alpha-SMA) and figure 6 (SMAD3), the changes show lot of variability.
Response: Thank you for your comment. We added the densitometries in Figure 5b and Figure 6b, respectively.
Comment 4. Lee YH and coll. investigate whether estrogen deficiency (ED) potentiates hepatic fibrosis in thioacetamide (TAA)-treated rat model, using biochemical, molecular and histopathological approaches. They conclude that ED potentiates TAA-induced oxidative damages in the liver, suggesting that estrogen deficiency may enhance the risk of hepatic fibrosis in menopausal women.
Response: Thank you for your critical comment. We clearly described this part and thus we revised again. Our results indicated that estrogen deficiency (ED) potentiate the severity of TAA-induced liver fibrosis as similar to that of observed in male. We revised this part “menopausal woman may increase the severity against chemical-induced hepatic damages or fibrosis compared with non-menopausal women”.
Comment 5. The aim of the present study is of a certain interest even if the role of estrogens in fibrosis, and in hepatic fibrosis too, is already extensively studied. Furthermore, the data presented are not convincing and further demonstrations should be provided.
Response: Thank you for your valuable comments. As you know, previous studies already indicated that estrogen play an important role in the prevention of hepatic diseases or fibrosis. As your comments, we revised and added “Although numerous investigators have suggested the protective effects of estrogen against hepatic fibrosis [23,36], the ED status may increase the severity of chemical-induced hepatic fibrosis are remaining unclear. Our findings clearly demonstrated that OVX female rats potentiated the severity of TAA-induced hepatic fibrosis in sham control rats. Further, the potency of hepatic damages exhibited in OVX rats were very similar as shown in male rats”.
Reviewer 2 Report
Lee YH and coll. investigate whether estrogen deficiency (ED) potentiates hepatic fibrosis in thioacetamide (TAA)-treated rat model, using biochemical, molecular and histopathological approaches. They conclude that ED potentiates TAA-induced oxidative damages in the liver, suggesting that estrogen deficiency may enhance the risk of hepatic fibrosis in menopausal women.
The aim of the present study is of a certain interest even if the role of estrogens in fibrosis, and in hepatic fibrosis too, is already extensively studied. Furthermore, the data presented are not convincing and further demonstrations should be provided.
Major concerns are raised that need to be clarified.
1) In the figure… the size of the portal tracts and central veins are not comparable in the different experimental settings. They seem to be constituted by vascular and biliary structures of different size and their enlargement could be not exclusively due to a different amount of collagen deposition as the authors suggest. Probably histological fields at a lesser magnification should be utilized in order to better visualize the fibrosis degree in OVX/TAA and TAA mice.
2) In the figure… an effect on apoptosis of ED was derived by the analysis of TUNEL reaction. However, the TUNEL-positive nuclei seem to belong mainly to non-parenchymal cells rather than hepatocytes, the main cell type in which apoptosis should be identified and quantified.
Author Response
Reviewer #2
Comment 1. In the figure… the size of the portal tracts and central veins are not comparable in the different experimental settings. They seem to be constituted by vascular and biliary structures of different size and their enlargement could be not exclusively due to a different amount of collagen deposition as the authors suggest. Probably histological fields at a lesser magnification should be utilized in order to better visualize the fibrosis degree in OVX/TAA and TAA mice.
Response: Thank you for your excellent comments. We changed Fig. 5c.
Comment 2. In the figure… an effect on apoptosis of ED was derived by the analysis of TUNEL reaction. However, the TUNEL-positive nuclei seem to belong mainly to non-parenchymal cells (NPC) rather than hepatocytes, the main cell type in which apoptosis should be identified and quantified.
Response: Thank you for your nice comment. We carefully checked and revised this part. TUNEL-positive apoptotic liver cells nuclei (hepatocytes and NPC) detected in the liver sections.
Round 2
Reviewer 1 Report
None
Author Response
None
Reviewer 2 Report
The authors improved the manuscript except for figure 2 and 4, in which further work could be done. I recommend to further revise requested figures.
In figure 2 portal tract or central vein of different size are presented.
In figure 4 no hepatocyte nucleus is indicated by arrows, by my opinion.
Author Response
Reviewer #2
The authors improved the manuscript except for figure 2 and 4, in which further work could be done. I recommend to further revise requested figures.
Comment 1. In figure 2 portal tract or central vein of different size are presented.
Response: Thank you for your comment. We changed Fig. 2c.
Comment 2. In figure 4 no hepatocyte nucleus is indicated by arrows, by my opinion.
Response: Thank you for your excellent comments. We changed Fig. 5c.
Round 3
Reviewer 2 Report
Authors must be acknowledged to have significantly improved the quality of data presentation.The manuscript in the present form is able to offer a significant contribution in the specific field, in my opinion.
Author Response
Dear Editor-in-Chief,
Thank you for your valuable comments.
As your comment, altered copper bioavailability predicts early atherosclerosis as main CV risk in obese patients with hepatic steatosis.
However, this comment seems to be irrelevant to our resrahch.
The aim of the present study is to investigate whether estrogen deficiency (ED) potentiates hepatic fibrosis. We did not considered the other factors uch endocrine disruptors (EDC) and heavy metals to evaluate hepatic fibrosis in the animal model (OVX model).
I'd like to thank you for your kind help.
Best wishes,